

# Comparative analysis of *Spodoptera frugiperda* (J. E. Smith) (Lepidoptera, Noctuidae) corn and rice strains microbiota revealed minor changes across life cycle and strain endosymbiont association

Sandra María Marulanda-Moreno[1,*], Clara Inés Saldamando-Benjumea[2], Rafael Vivero Gomez[3], Gloria Cadavid-Restrepo[1] and Claudia Ximena Moreno-Herrera[1,*]

[1] Grupo de Microbiodiversidad y Bioprospección-Microbiop, Departamento de Biociencias, Facultad de Ciencias, Universidad Nacional de Colombia, sede Medellín, Colombia
[2] Grupo de Biotecnología Vegetal UNALMED-CIB. Línea en Ecología y Evolución de Insectos, Facultad de Ciencias, Universidad Nacional de Colombia, Medellín, Colombia
[3] Grupo de Microbiodiversidad y Bioprospección-Microbiop, Universidad Nacional de Colombia, sede Medellín, Colombia
* These authors contributed equally to this work.

Corresponding author
Claudia Ximena Moreno-Herrera, cxmoreno@unal.edu.co

## ABSTRACT

**Background:** *Spodoptera frugiperda* (FAW) is a pest that poses a significant threat to corn production worldwide, causing millions of dollars in losses. The species has evolved into two strains (corn and rice) that differ in their genetics, reproductive isolation, and resistance to insecticides and *Bacillus thuringiensis* endotoxins. The microbiota plays an important role in insects' physiology, nutrient acquisition, and response to chemical and biological controls. Several studies have been carried out on FAW microbiota from larvae guts using laboratory or field samples and a couple of studies have analyzed the corn strain microbiota across its life cycle. This investigation reveals the first comparison between corn strain (CS) and rice strain (RS) of FAW during different developmental insect stages and, more importantly, endosymbiont detection in both strains, highlighting the importance of studying both FAW populations and samples from different stages.
**Methods:** The composition of microbiota during the life cycle of the FAW corn and rice strains was analyzed through high-throughput sequencing of the bacterial 16S rRNA gene using the MiSeq system. Additionally, culture-dependent techniques were used to isolate gut bacteria and the Transcribed Internal Spacer-ITS, 16S rRNA, and *gyr*B genes were examined to enhance bacterial identification.
**Results:** Richness, diversity, and bacterial composition changed significantly across the life cycle of FAW. Most diversity was observed in eggs and males. Differences in gut microbiota diversity between CS and RS were minor. However, *Leuconostoc*, A2, *Klebsiella*, *Lachnoclostridium*, *Spiroplasma*, and *Mucispirilum* were mainly associated with RS and *Colidextribacter*, *Pelomonas*, *Weissella*, and *Arsenophonus* to CS, suggesting that FAW strains differ in several genera according to the host plant. Firmicutes and Proteobacteria were the dominant phyla during FAW

metamorphosis. *Illeobacterium, Ralstonia*, and *Burkholderia* exhibited similar abundances in both strains. *Enterococcus* was identified as a conserved taxon across the entire FAW life cycle. Microbiota core communities mainly consisted of *Enterococcus* and *Illeobacterium*. A positive correlation was found between *Spiroplasma* with RS (sampled from eggs, larvae, pupae, and adults) and *Arsenophonus* (sampled from eggs, larvae, and adults) with CS. *Enterococcus mundtii* was predominant in all developmental stages. Previous studies have suggested its importance in FAW response to *B. thuringensis*. Our results are relevant for the characterization of FAW corn and rice strains microbiota to develop new strategies for their control. Detection of *Arsenophonus* in CS and *Spiroplasma* in RS are promising for the improvement of this pest management, as these bacteria induce male killing and larvae fitness reduction in other Lepidoptera species.

## INTRODUCTION

*Spodoptera frugiperda*, also known as fall armyworm (FAW), is an endemic species of the Western Hemisphere and one of the most devastating pests in corn worldwide (*Yainna et al., 2022*; *Guo et al., 2022*; *Fu et al., 2023*; *Tambo et al., 2023*). Currently, FAW has been characterized as an invasive species in several African countries (*Yainna et al., 2022*), India (*Sharanabasappa et al., 2018*), Thailand (*IPPC, 2018*), China (*IPPC, 2019*) and Australia (*FAO, 2020*). According to *Tambo et al. (2023)*, the FAW infestations can result in corn production losses of 4.1 to 17.7 million tons annually, which is valued at US$1.1–4.7 billion in just twelve maize-producing countries. FAW can be found over 350 host plants (*Montezano et al., 2018*), including rice, sorghum, sugar cane, cotton, peanuts, soybeans, and alfalfa, among others (*Atnafu et al., 2021*; *Chhetri & Acharya, 2019*; *Guo et al., 2022*; *Wang et al., 2023*), producing losses from 15% to 75% in Latin America (*Ayil-Gutiérrez et al., 2018*). In Colombia, FAW is the most damaging pest in corn, followed by rice, cotton, sorghum, and sugar cane (*Cano-Calle, Arango-Isaza & Saldamando-Benjumea, 2015*).

FAW has evolved into two populations that differ in their genetics, named the corn strain (CS) and the rice strain (RS) (*Pashley-Prowell, McMichael & Silvain, 2004*). Both strains are molecularly identified with several markers, including a PCR-RFLP of the mitochondrial gene *COI*, a PCR of the nuclear gene FR, and sequencing of the triose phosphate isomerase (*Tpi*) gene (*Nagoshi et al., 2020*). A genome comparison made between FAW corn populations from Mississippi (susceptible) and Puerto Rico (resistant) showed that the copy number variation of the detoxification genes to deltamethrin was higher in the later population, suggesting that the species rapidly adapts to this selection pressure (*Gimenez et al., 2020*). FAW Colombian strains exhibit differences in resistance to chemical and biological controls in Central Colombia. *Ríos-Díez, Siegfried & Saldamando-Benjumea (2012)* demonstrated that the CS is more resistant than the RS to

*Bacillus thuringiensis* (*Bt*) endotoxins (Cry1AC and Cry1AB) in laboratory conditions. Also, *Ríos-Díez & Saldamando-Benjumea (2011)* found that the RS is more resistant to insecticides (lambda-cyhalothrin and methomyl) than CS.

FAW is known to have genetic resistance to insecticides and Bt endotoxin (*Yainna et al., 2022*; *Fiteni et al., 2022*; *Gimenez et al., 2020*); however, the swift development of FAW's resistance to insecticides and Bt (*Gimenez et al., 2020*; *Castañeda-Molina et al., 2023*) highlights the need for finding more efficient ways to control it. The effects of microbiota on host insects have now provided new perspectives for the development of alternative strategies for pest control (*Fu et al., 2023*). The insect microbiota is associated with a wide range of beneficial host–microbe functions, including food digestion, nutrition, protection, immune response modulation, and detoxification of xenobiotics (*Almeida et al., 2017*; *Guo et al., 2022*; *Jeon et al., 2023*). In a study conducted in Colombia, two species of *Enterococcus* (*E. mundtii* and *E. casseliflavus*) were found in the gut microbiota of the corn strain FAW larvae. These species were resistant to antibiotics and persisted in the presence of Bt endotoxins. Additionally, the endosymbiont *Arsenophonus* sp. was detected in the gonads of both male and female FAWs using specific primers (*Castañeda-Molina et al., 2023*). In Brazil, *Almeida et al. (2017)* identified *Enterococcus mundtii, Enterococcus casselflavus, Delftia lacustris, Leclercia adecarboxylata, Microbacterium paraoxydans, Pseudomonas stutzeri, Arthrobacter nicotinovorans, Pseudomonas psychrotolerans, Microbacterium arborescens,* and *Staphylococcus sciuri* subsp. *sciuri* in the gut of pesticide-resistant FAW. These bacteria can tolerate and break down several insecticides, such as deltamethrin, lambda-cyhalothrin, chlorpyrifos ethyl, spinosad, and lufenuron, as observed in laboratory experiments. The microbiota isolated from insects through culture-dependent approaches is also essential for understanding the physiological potential of isolated organisms and can be used in other experiments (*Higuita-Palacio et al., 2021*), particularly from insect's life cycles (*Chen et al., 2016*; *Li et al., 2022*; *Fu et al., 2023*) as they can be tested against *Bt* endotoxins (*Castañeda-Molina et al., 2023*) or insecticides (*Almeida et al., 2017*; *Gomes, Omoto & Cônsoli, 2020*), even though their diversity is higher from field samples (*Gomes, Omoto & Cônsoli, 2020*) as they have to be kept under laboratory conditions. The discovery of endosymbionts can also be crucial in finding new strategies for insect control. For example, in *Aedes aegypti*, the bacterium *Wolbachia* sp. is involved in male genitalia changes into female genitalia and cytoplasmic incompatibilities (*Zhang & Lui, 2020*). Also, *Arsenophonus nasoniae* produces male killing in *Nasonia* sp. wasp (*Taylor et al., 2011*).

Several studies have been conducted to investigate the gut microbiota of the fall armyworm (FAW), using both field and laboratory populations, as well as insects fed on different diets (*Jeon et al., 2023*; *Jones et al., 2019*; *Oliveira & Cônsoli, 2023*; *Mason et al., 2020*; *Xu et al., 2022*). These studies have provided a single snapshot of the microbial species composition of the community (*Chen et al., 2016*). Other studies have analyzed the microbiota in the corn strain during the FAW life cycle (*Fu et al., 2023*; *Li et al., 2022*; *Lü et al., 2023*). Nevertheless, until now, none of these works has comprehensively investigated the FAW (CS and RS) strain microbiota throughout their life cycle. The results of this study are valuable as they provide additional information on microbiota, specifically

regarding endosymbiont identification in corn and rice strains. This information can be used to improve the management of FAW strains.

## MATERIALS AND METHODS

### Ethics statement

The collection of the larvae and genetic access was provided by ANLA (Autoridad Nacional de Licencias Ambientales) to Universidad Nacional de Colombia. Framework permission collection of wild specimens, resolution 0255, 03/14/2014 (article 3).

### Insect collection and breeding

FAW larvae (third to the sixth instar, $N = 250$) were collected from corn and rice fields. Sampling from corn was performed at Estación Agraria Cotové in the municipality of Santa Fé de Antioquia (Universidad Nacional de Colombia, Medellín campus) (6°31′54.0 ″ N 75° 49′33.8″ W) during June 2019. Sampling from rice was done in October 2019 at the municipality of El Espinal (4°08′55″N 74°52′55″W) in Tolima's department. The collected larvae were kept in separate containers and taken to the Laboratorio de Ecología y Evolución de Insectos at Universidad Nacional de Colombia, Medellín campus. They were fed with corn and rice leaves until they became pupae. The adults were fed with a sterile water-honey solution (1:1 ratio) using a cotton ball, and their food was changed daily. Based on genotypification, two colonies, named corn strain (CS) and rice strain (RS), were kept in the laboratory. The first generation (F1 generation) of eggs, larvae, pupae, and adults (males and females) were processed for subsequent microbiota analyses. The larvae obtained from this generation were fed with a sterile bean diet according to the modified protocol of *Shorey & Hale (1965)*. Insects were kept at 25 °C ±2.70% ±5 RH (relative humidity) and a photoperiod of 12:12 h–L: D (light: dark).

### Identification of FAW strains

DNA was extracted from the cephalic and caudal regions of larvae and adult specimens of the Fall Armyworm (FAW). Furthermore, DNA was also extracted from a portion of the pupal and egg tissues using the grind buffer method as described by *Porter & Collins (1991)*. PCR-RFLP of the mitochondrial cytochrome oxidase subunit I (COI) gene and a PCR of the nuclear gene FR (For Rice) were used to identify the CS and RS populations based on *Cano-Calle, Arango-Isaza & Saldamando-Benjumea (2015)* protocol.

### Culture-independent method

#### Processing of FAW for microbiota analysis

DNA was extracted from three to five egg masses per FAW strain by placing them into 1.5 mL Eppendorf tubes. Larvae of corn and rice strains were separated into two groups, early instar (Larvae L1-L3, $N = 6$ larvae/strain) and late instar (Larvae L4-L6, $N = 6$ larvae/strain), and placed into separate 1.5 mL Eppendorf tubes. The corn and rice strain pupae ($N = 6$/strain) were also separated, as well as their adults ($N = 6$ per sex). Each sample was washed separately with a 1X phosphate-buffered saline solution (PBS, pH = 7.4) (Life
Technologies Corp., Frederick, MD, USA) +1% Tween-20 (1 min) followed by ethanol disinfection at 70% (1 min) and a final rinse with PBS 1× (1 min) (*Higuita-Palacio et al., 2021*). The complete gastrointestinal tract of each individual was dissected using sterile forceps (Figs. S1A and S1B). The pupae were washed and disinfected following the protocol previously described by *Higuita-Palacio et al. (2021)* and deposited in pools of six individuals in 1.5 mL Eppendorf tubes. All tissue samples were weighed and macerated in 500 μL of sterile 1X PBS. A total of 100 μL of the homogenate was used to study bacterial communities by culture-dependent methods; the remaining tissue was preserved in 70% ethanol at −20 °C for culture-independent analyses. All procedures were performed under sterile conditions.

### DNA extraction and microbiome composition by high-throughput sequencing of 16S rRNA gene

DNA was extracted from different CS and RS instars using the Fast DNA Kit by MP Biomedicals, according to the manufacturer's recommendations. The DNA samples were assessed for concentration (Table 1), quality, and purity using a Thermo scientific ND-100 Nanodrop spectrophotometer (Thermo Fisher Scientific, MA, USA) and visualization on 1% agarose gels stained with E-ZVision dye. For DNA analysis, 28 samples (18 of CS and 10 of RS) were selected, and the 16S rRNA gene V4 hypervariable region was sequenced using the Illumina MiSeq platform.

### Statistical and bioinformatics analysis

The DADA2 program (https://benjjneb.github.io/dada2/) was used to remove DNA chimeras, filter low-quality sequences, and assemble the amplicons. The amplicon sequence variants (ASVs) produced by this program were obtained and then aligned against the Ribosomal Data Project (RDP) database. ASVs with a minimum frequency of 0.05% per sample were chosen. The Microbiome Analyst tool (https://www.microbiomeanalyst.ca/) (*R Core Team, 2022*) was used for the analysis of bacterial communities. The rarefaction curves were normalized concerning the sample with the lowest number of reads. The normalized data was employed for the microbiota diversity analyses. The alpha diversity obtained per instar and FAW strain was estimated by measuring the Shannon-Wiener, Simpson, and Chao1 indexes. Further on, a principal coordinate analysis (PCoA) was performed using the Bray-Curtis distance matrix to compare the structure of bacterial communities during FAW insect life cycle instars and between CS and RS strains. The permutational analysis of variance (PERMANOVA) was used to test for statistical differences among samples based on the Bray-Curtis distance. The hierarchical clustering analysis was also performed based on the Pearson correlation coefficient. In addition, a linear discriminant analysis (LDA) (LEfSe) (*Segata et al., 2011*) was used to identify bacterial ASVs that significantly differ between FAW strains or developmental stages. Finally, a microbial core analysis was obtained from FAW strains and all instars tested, considering the genera in 50% or more of the samples and with a relative abundance >0.02% in each library.

**Table 1 List of samples of FAW of different CS and RS instars processed to obtain total DNA samples analyzed by high-throughput sequencing of 16S rRNA gene.**

| Origin | Sample code | Short code | Stage | Concentration (dry ng) |
|--------|-------------|------------|-------|-----------------------:|
| Corn   | Eggs_C1     | CE1        | Eggs        | 1,340  |
|        | Eggs_C2     | CE2        | Eggs        | 936    |
|        | Eggs_C3     | CE3        | Eggs        | 1,436  |
|        | Eggs_C4     | CE4        | Eggs        | 4,396  |
|        | Larvae_Y_C2 | CYL2       | Larvae L1-3 | 696    |
|        | Larvae_L_C1 | CLL1       | Larvae L4-6 | 656    |
|        | Larvae_L_C2 | CLL2       | Larvae L4-6 | 1,435  |
|        | Larvae_L_C3 | CLL3       | Larvae L4-6 | 4,188  |
|        | Pupae_C2    | CP2        | Pupae       | 11,501 |
|        | Pupae_C3    | CP3        | Pupae       | 455    |
|        | Pupae_C5    | CP5        | Pupae       | 1,930  |
|        | Pupae_C6    | CP6        | Pupae       | 1,975  |
|        | Pupae_C7    | CP7        | Pupae       | 2,185  |
|        | Adult_M_C3  | CMA3       | Adult male  | 981    |
|        | Adult_M_C5  | CMA5       | Adult male  | 1,456  |
|        | Adult_F_C1  | CFA1       | Adult female| 835    |
|        | Adult_F_C2  | CFA2       | Adult female| 960    |
|        | Adult_F_C3  | CFA3       | Adult female| 780    |
| Rice   | Eggs_R1     | RE1        | Eggs        | 2,028  |
|        | Eggs_R2     | RE2        | Eggs        | 1,496  |
|        | Eggs_R3     | RE3        | Eggs        | 4,108  |
|        | Larvae_Y_R1 | RYL1       | Larvae L1-3 | 776    |
|        | Larvae_Y_R2 | RYL2       | Larvae L1-3 | 976    |
|        | Larvae_L_R1 | RLL1       | Larvae L4-6 | 3,752  |
|        | Larvae_L_R2 | RLL2       | Larvae L4-6 | 3,092  |
|        | Larvae_L_R3 | RLL3       | Larvae L4-6 | 1,435  |
|        | Pupae_R1    | RP1        | Pupae       | 5,622  |
|        | Pupae_R2    | RP2        | Pupae       | 495    |

## Culture-dependent methods

### Isolation and colony-forming units of microbiota from FAW

Serial dilutions were made from each tissue homogenate of CS and RS of FAW in sterile PBS, up to a dilution factor of 1:10−7. To estimate the number of cultivable bacteria, standard agar plate-based cultivation was employed on Luria Bertani (LB) agar. Each assay was performed in triplicate, and cultures were incubated at 37 °C for 24 to 48 h. Colony forming units (CFU)/g were calculated by counting the colonies on plates with 30 to 300 colonies. The data were transformed into a log of CFU/g. The statistical differences in the total cultivable bacteria amongst samples were estimated using Krustal Wallis and Dunn's tests in the program SPSS Statistics® version 22 (*IBM Corp, 2013*). The bacterial morphotypes were identified based on their macroscopic characteristics, such as shape,

edge, color, and elevation. Each morphotype was purified by successive passages on LB agar and characterized microscopically through Gram staining.

### Extraction of genomic DNA and PCR of the transcribed internal spacer-ITS, 16S rRNA, and gyrB genes

Bacterial DNA was extracted using the thermal lysis. The universal primers L1 (5′ CAAGGCATCCACCGTGT3′) and G1 (5′GAAGTCGTAACAAGG3′) were used for PCR amplification of the internal transcribed spacer (RISA) (*Jensen, Webster & Straus, 1993*). PCR products were separated on (8% polyacrylamide) (PAGE) gels and visualized by staining with silver nitrate (AgNO₃) (Amresco, Solon, Ohio). Cluster analysis was performed using the Pearson correlation coefficient and the single-linked clustering method in the GelCompar II software (Applied Maths Biosystems, Kortrijk, Belgium). Representative isolates from each cluster with a percentage similarity ≥65% were chosen for further molecular identification. The identification of the selected bacteria was carried out through PCR of the 16S rRNA gene, using the Eubac 27F (5′ AGAGTTTGATCCTGGCTCAG3′) and 1492R (5′GGT TACCTT GTT ACG ACT T3′) primers (*Espejo et al., 1998*). To confirm the species of some of the isolates, a region of approximately 1,260 bp of the *gyr*B gene was amplified using the UP1 primers (5′–AGC AGG GTA CGG ATG TGC GAG CCR TCN ACR TCN GCR TCN GTC AT–3′) and UP2R (5′–GAA GTC ATC ATG ACC GTT CTG CAY GCN GGN GGN AAR TTY GA–3′) (*Yamamoto & Harayama, 1995*) according to the protocol described by *Cano-Calle et al. (2022)*. The PCR products of 16S rDNA and the *gyr*B gene were analyzed by electrophoresis in 1.2% agarose gel stained with the fluorescent marker EZ.VisionTM (Amresco, U.S.A). Once the reactions were confirmed, the PCR products were sequenced on an ABI PRISM 3100 Genetic Analyzer (Applied Biosystems, Carlsbad, CA, USA).

The Geneious Prime 2021 software (version 2.2) was used to edit and generate consensus sequences. The cured and assembled sequences were compared with the GenBank database using the BLASTn algorithm (National Center for Biotechnology Information; http://www.ncbi.nlm.nih.gov/BLAST/). Sequence alignments were obtained using the ClustalW tool (*Thompson, Higgins & Gibson, 1994*) found in the software Mega X (*Kumar et al., 2018*). Bayesian dendrograms were built using Beast v2.6.7 (*Drummond & Rambaut, 2007*). The generated sequences can be consulted in the NCBI GenBank database with accession numbers CFA1C1(OR546283) to RMA1C2 (OR546319), and for *gyr*B gene are CYL2C1 (OR592215) to CE1C6 (OR592219).

## RESULTS

### Identification of CS and RS FAW strains

The analysis of the COI region of mitochondrial DNA using the PCR-RFLP technique showed that the 18 samples of FAW collected from corn plants belonged to the CS as they were digested with *Msp*I restriction enzyme (Figs. S2–S2A). On the other hand, all 10 FAW samples obtained from rice plants remained undigested, which is consistent with the RS (Figs. S2–S2B). The PCR of the nuclear FR gene exhibited a smear pattern of 500 bp in only

the ten individuals that were collected in rice crops. Hybrids between FAW strains were not detected (Figs. S2–S2C) (*Cano-Calle, Arango-Isaza & Saldamando-Benjumea, 2015*).

## Microbiome composition by 16S rRNA gene by high-throughput sequencing

A total of 7.674.095 assigned reads in 7.987 ASVs with an identity greater than 97% were obtained from all CS and RS instar samples. After the removal of chimeric sequences, ASVs with abundances less than 0.0001% were selected and 4.023.896 reads remained with an average of 143.710 reads per sample. The number of sequences per sample was normalized to 45.019 before further analyses. The rarefaction analysis showed an appropriate depth per sample to the number of reads for the alpha and beta diversity analyses (Fig. S3). All samples exhibited a Good's coverage index of over 99.9, which indicates that the sequencing quality was good and that there was adequate representation of the species in all samples. After the quality filter, the reads were categorized into nine phyla, 14 classes, 33 orders, 54 families, and 81 bacterial genera.

## Diversity of bacterial communities in strains and developmental stages of FAW

The richness and evenness of bacterial species were not significantly different between FAW strains according to the alpha diversity metrics (Chao1, Shannon, and Simpson indices) (Figs. 1A–1C). Likewise, the ß-diversity analysis showed no significant effect on bacterial diversity according to FAW strain. Also, no clustering per strain was found from the Principal Coordinate Analysis (PCoA) based on the Bray-Curtis dissimilarity index ((PERMANOVA) F-value = 0.076874; R-squared = 0.002948; $p$-value < 0.977) (Fig. 2A).

During the development of the fall armyworm the highest bacterial diversity was detected in the eggs. As the FAW progressed through its lifecycle, there was a gradual reduction in species richness from larvae to pupae (Figs. 1D–1F). Females had similar richness values to the pupae, while the males and young larvae exhibited the highest richness following the egg's richness. Shannon and Simpson diversity estimators showed a similar trend throughout FAW metamorphosis; however, the females had the lowest diversity compared to the other developmental stages. There were significant differences in alpha diversities during the development stages of FAW, as shown in Figs. 1D–1F. The principal coordinate analysis (PCoA) using the Bray-Curtis dissimilarity index showed a variation in bacterial diversity in the FAW developmental stages. Clustering amongst eggs, young larvae, and male microbiota was separate from another clustering composed of older larvae, pupae, and females The PERMANOVA test further confirmed these results as it demonstrated significant differences amongst FAW bacterial communities at different developmental stages ((PERMANOVA) F-value = 6.9355; R-squared = 0.61184; $p$-value < 0.001) (Fig. 2B).

## Microbiota composition associated with the FAW

Taxonomic analysis showed that Firmicutes and Proteobacteria were the most common phyla found in all developmental stages of FAW. The relative abundance of Firmicutes

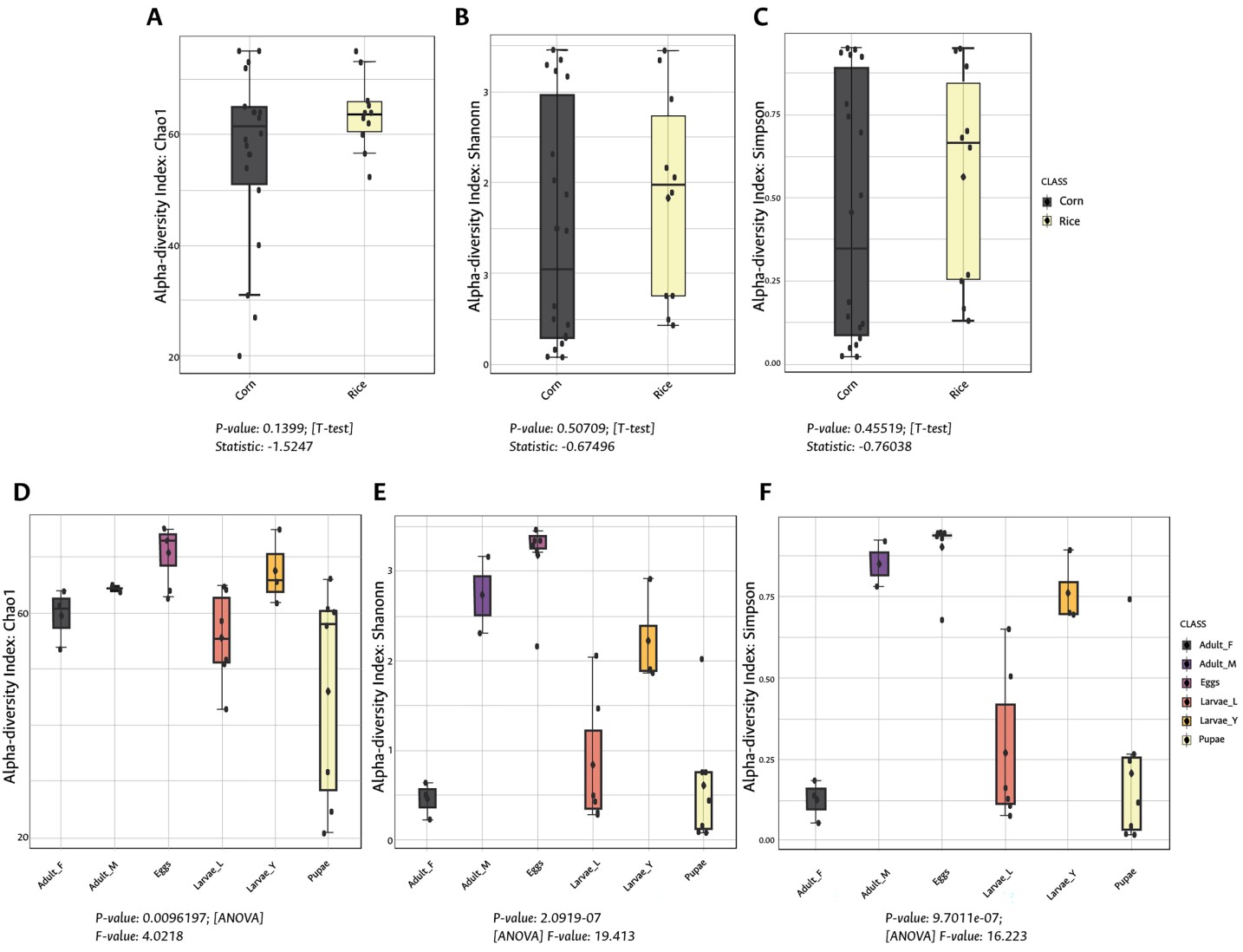

**Figure 1 Alpha diversity (intra-diversity) of bacterial composition in corn and rice strains (A–C) and stages of FAW life cycle (D–F).** A higher value in each index indicates a richer and more diverse bacterial composition. Corn and Rice, strain of FAW; Adult_F, adult females; Adult_M, adult males; Larvae_Y, young larvae (L1-L3 larvae); Larvae_L, late larvae (L4-L6 larvae). The statistical values from the Test t (pairwise comparison) and ANOVA (group comparison) are shown in which box.

increased from 48% in the eggs, to 74% in L1-3 larvae, and to 92% and 95% in L4-6 larvae and pupae, respectively. In females, the microbiota was predominantly composed of this phylum (96%), whereas in males, it significantly decreased (56%). In contrast, a reduction of the phylum Proteobacteria was detected in FAW, as its relative abundance was reduced from 22% in the eggs, 14% in the L1-3 larvae, 5% in the L4-6 larvae, and 2% in the pupae. It is important to mention that the phylum Proteobacteria is almost absent in males as they have a relative abundance of 1%, while in females, the abundance is 23% (Fig. 3A).

At the genus level, the egg microbiota was mainly composed of *Ileibacterium*, *Dubosiella*, *Enterococcus*, *Turicibacter*, *Ralstonia*, and *Burkholderia*. These genera were detected in all samples but were more abundant in the eggs, followed by the L1-3 larvae

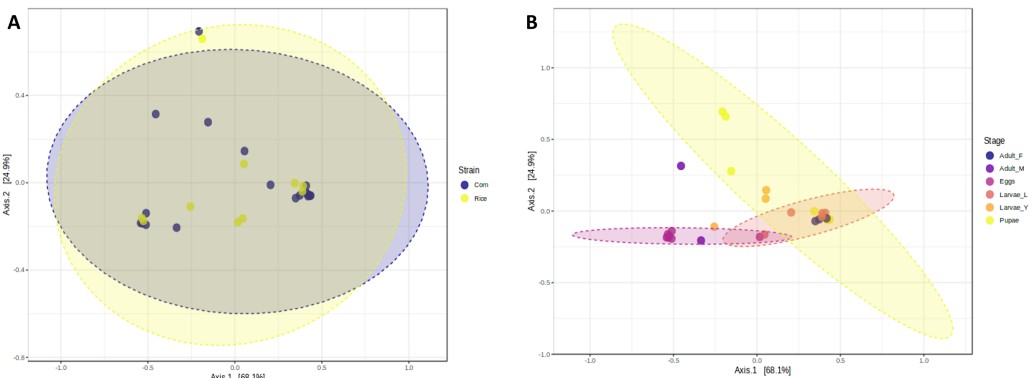

**Figure 2 Principal coordinates analysis (PCoA) based on the Bray-Curtis similarity of bacterial communities in corn and rice strains (A) and different stages of FAW life cycle (B).**

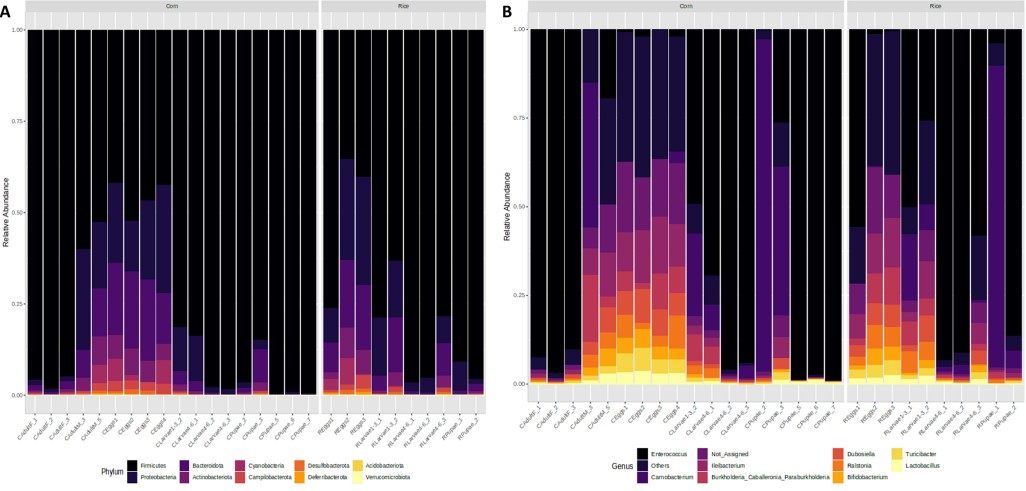

**Figure 3 Microbiota composition associated with corn and rice strains and stages of FAW life cycle.**
Relative abundance of ASVs that were called to the taxonomic rank of phylum (A) and genus (B). The ASVs with abundances above 1% of the rarefied count matrix were selected. NA: ASVs without taxonomic assignment.

and the males, and had lower abundances in the L4-6 larvae, the pupae, and the females (Fig. 3B).

The genus *Enterococcus* was found to be the most common in all developmental stages of the FAW, except in the eggs and adult males. The relative abundance of *Enterococcus* in eggs was 8.9%. However, after hatching, the abundance of *Enterococcus* increased dramatically from 41.6% in L1-3 larvae to 83.6% in L4-6 larvae. In pupae, the average relative abundance of *Enterococcus* was 58.8%. Upon completion of metamorphosis, females showed a high predominance of this genus (93.2%), while males showed only 9.8% of the microbiota to be *Enterococcus* (Fig. S4). The microbial core of all the samples at level genus was composed by *Enterococcus* and *Ileibacterium* (Fig. S5). Subsequently, LEfSe analysis identified a group of bacterial taxa that differentiated in microbiota abundance throughout the FAW life cycle. *Ileibacterium, Dubosiella, Ralstonia, Turicibacter*, and

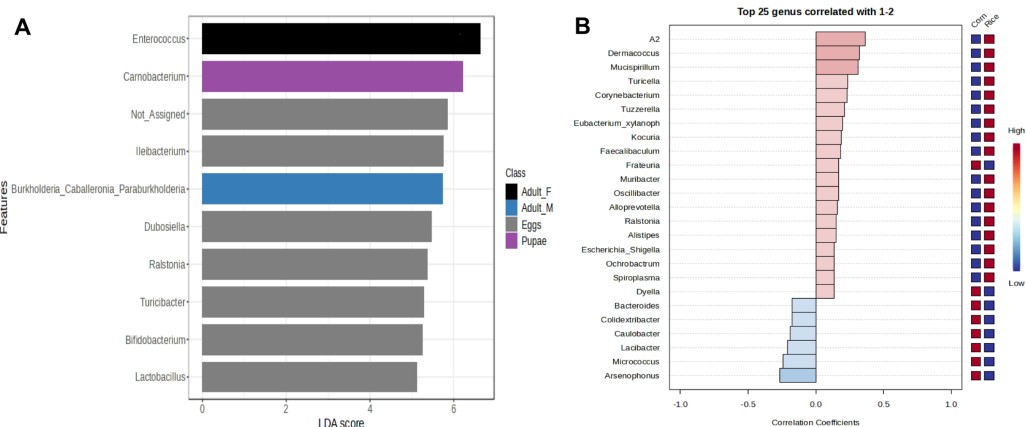

**Figure 4 Linear Discriminant Analysis Effect Size (LEfSe) of microbiota in stages of FAW life cycle (A). Patterns correlation analysis of FAW bacteria at the genus (B).** The top 25 genera correlated with corn and rice strains are shown. Red indicates positive correlation and blue negative correlation

*Bifidobacterium* were differentially abundant in eggs, while *Enterococcus, Burkholderia,* and *Carnobacterium* were differentially abundant in adult females, adult males, and pupae, respectively (Fig. 4A).

In addition, LEfSe analysis failed to detect bacterial taxa that differ in abundance between FAW strains. However, the results of the correlation analysis, showed an association of the genera *Leuconostoc,* A2, *Klebsiella, Lachnoclostridium, Spiroplasma,* and *Mucispirilum* with RS; and an association of the genera *Colidextribacter, Pelomonas, Weissella,* and *Arsenophonus* with CS, suggesting that FAW strains differ in several genera according to host plant (Fig. 4B).

## Isolation and culture of microbial isolates from FAW

The egg samples had the lowest CFU/g counts, with 5.8 and 6.2 log CFU/g for the RS and CS, respectively. In contrast, the highest counts were observed in the CS young larvae with counts of 9.4 log CFU/g and in RS pupae with 9.7 log CFU/g (Fig. S6). The CFU/g counts significantly differed among different FAW developmental stages (Kruskal-Wallis $p < 0.05$). 87 bacterial isolates (44 CS and 43 RS) were recovered and identified according to their macroscopic and microscopic characteristics, with 92% of the isolates being gram-positive bacteria and the remaining 8% being gram-negative bacteria. These isolates were also molecularly identified using the internal transcribed spacer region (ITS) (Fig. S7).

## Molecular characterization and identification of bacterial isolates

Clustering analysis, based on a 65% shared similarity value among samples, was conducted using the Internal Transcribed Spacer Region (ITS), which differentiated twelve groups (Fig. S7). Further molecular identification was carried out on 37 representative bacteria strains. The analysis of partial sequences of the 16S rDNA revealed that the culturable bacteria community of FAW mainly consisted of members of the Firmicutes and Proteobacteria phyla (Table 2 and Fig. 5). Two genera of the phylum Actinobacteria were

**Table 2 Taxonomic identification of gut bacterial isolates of corn and rice strains of FAW according to their similarity to 16S rDNA gene sequences.**

| Strain | Isolated code | NCBI-GenBank Accession number | Stage | Phylogenetic affiliation | Phylum | Similarity (%) |
|--------|--------------|-------------------------------|-------|--------------------------|--------|----------------|
| Corn | CFA1C1 | OR546283 | Female adult | *Enterococcus mundtii* | Firmicutes | 99.4 |
| | CFA2C2 | OR546284 | | *Enterococcus haemoperoxidus* | | 99.6 |
| | CE1C1 | OR546285 | Eggs | *Enterococcus mundtii* | Firmicutes | 99.7 |
| | CE4C3 | OR546286 | | *Pediococcus pentosaceus* | | 99.7 |
| | CE1C4 | OR546287 | | *Bacillus amyloliquefaciens* | | 99.2 |
| | CE1C5 | OR546288 | | *Staphylococcus warneri* | | 99.7 |
| | CE1C6 | OR546289 | | *Bacillus pumilus* | | 99.5 |
| | CE2C4 | OR546290 | | *Staphylococcus saccharolyticus* | | 99.6 |
| | CE2C1 | OR546291 | | *Staphylococcus saccharolyticus* | | 99.8 |
| | CE2C6 | OR546292 | | *Cellulomonas pakistanensis* | Actinobacteria | 99.3 |
| | CE1C3 | OR546293 | | *Leclercia adecarboxylata* | Proteobacteria | 99.7 |
| | CYL2C4 | OR546294 | Young larvae | *Paenibacillus urinalis* | Firmicutes | 99.1 |
| | CYL1C2 | OR546295 | | *Enterococcus casseliflavus* | | 97.7 |
| | CYL2C1 | OR546296 | | *Enterococcus mundtii* | | 99.1 |
| | CYL2C2 | OR546297 | | *Enterococcus casseliflavus* | | 98.8 |
| | CLL1C1 | OR546298 | Late larvae | *Enterococcus mundtii strain* | Firmicutes | 100.0 |
| | CLL1C5 | OR546299 | | *Enterococcus casseliflavus* | | 100.0 |
| | CLL2C2 | OR546300 | | *Enterococcus casseliflavus* | | 99.7 |
| | CLL1C8 | OR546301 | | *Curtobacterium oceanosedimentum* | Actinobacteria | 99.7 |
| | CP6C7 | OR546302 | Pupae | *Lysinibacillus chungkukjangi* | Firmicutes | 100.0 |
| | CP1C2 | OR546303 | | *Enterococcus mundtii* | | 99.6 |
| | CMA5C2 | OR546304 | Male adult | *Enterococcus haemoperoxidus* | Firmicutes | 98.7 |
| | CMA5C4 | OR546305 | | *Terribacillus halophilus* | | 99.3 |
| Rice | RE1C4 | OR546306 | Eggs | *Staphylococcus capitis* | Firmicutes | 99.7 |
| | RYL1C2 | OR546307 | Young larvae | *Carnobacterium maltaromaticum* | Firmicutes | 96.9 |
| | RYL1C3 | OR546308 | | *Enterococcus casseliflavus* | | 99.9 |
| | RYL2C5 | OR546309 | | *Staphylococcus hominis subsp. novobiosepticus* | | 99.8 |
| | RLL1C6 | OR546310 | Late larvae | *Staphylococcus pasteuri* | Firmicutes | 99.8 |
| | RLL1C1 | OR546311 | | *Enterococcus mundtii* | | 98.8 |
| | RLL1C4 | OR546312 | | *Enterococcus casseliflavus* | | 99.8 |
| | RLL1C7 | OR546313 | | *Enterobacter tabaci* | Proteobacteria | 99.4 |
| | RLL2C5 | OR546314 | | *Enterobacter tabaci* | | 98.5 |
| | RP1C1 | OR546315 | Pupae | *Enterococcus mundtii* | Firmicutes | 99.9 |
| | RE3C2 | OR546316 | | *Planococcus massiliensis* | | 98.3 |
| | RP1C3 | OR546317 | | *Bacillus infantis* | | 99.7 |
| | RP3C1 | OR546318 | | *Enterococcus mundtii* | | 99.8 |
| | RMA1C2 | OR546319 | Male adult | *Enterococcus mundtii* | Firmicutes | 100.0 |

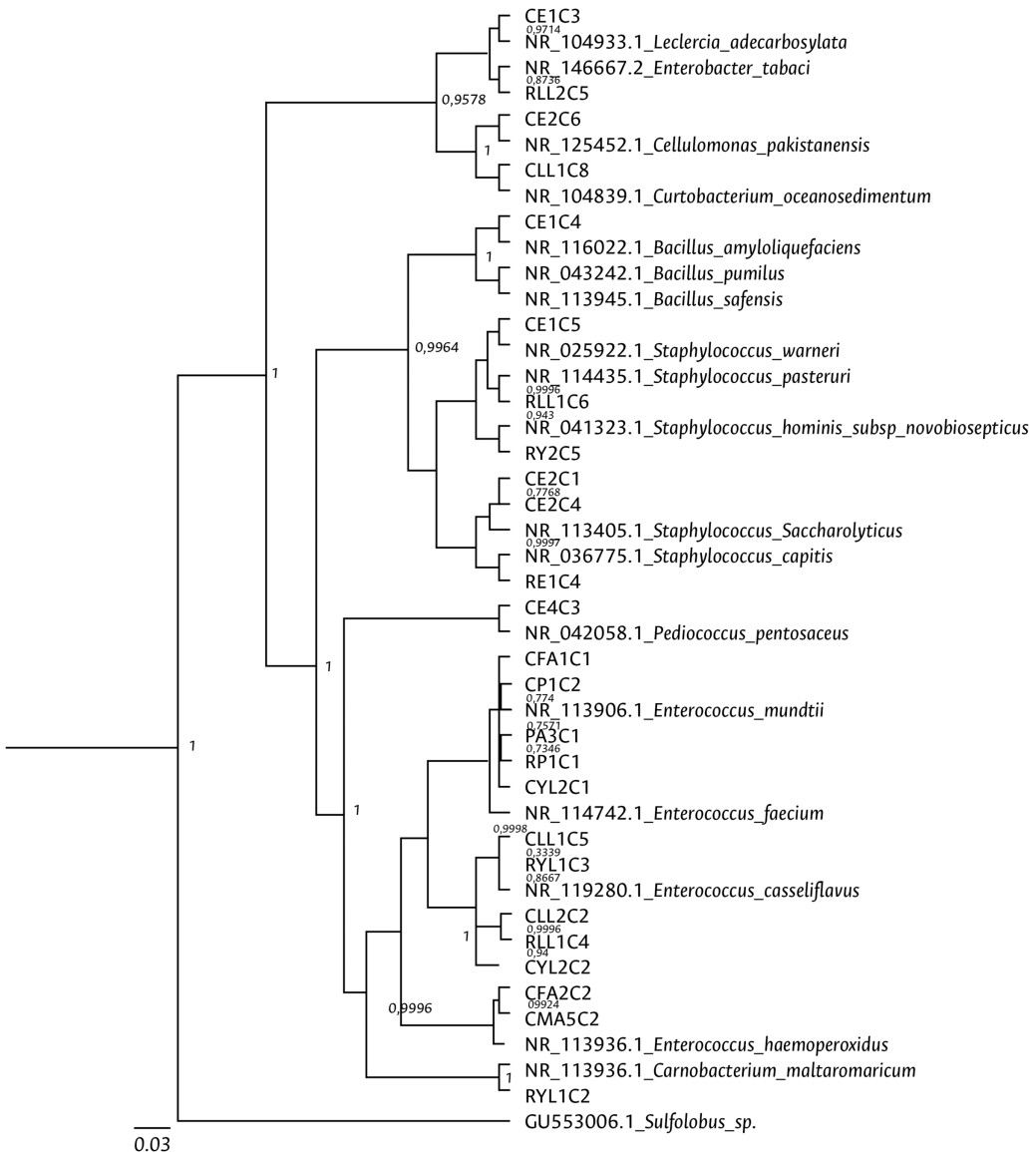

**Figure 5 Bayesian inference dendrogram for partial 16S rDNA sequences of bacteria isolated from FAW gut samples.** HKY (Hasegawa, Kishino and Yano) nucleotide substitution model (*Hasegawa, Kishino & Yano, 1985*) was used in the Beast program for the dendrogram construction. Bacterial isolates are represented with the assigned codes. The GenBank sequences are marked with the accession and identity code. The numbers in the nodes correspond to the posterior probability.

also isolated from eggs and guts of CS larvae. The genus *Enterococcus* was predominant in both FAW strains and was present in all CS and RS developmental stages. The species found in this genus were *E. casseliflavus*, *E. haemoperoxidus*, and *E. mundtii*. Three isolates identified as *E. mundtii* through the 16S rDNA molecular marker were further confirmed using the *gyr*B gene sequencing (Table 3 and Fig. 6).

**Table 3 Taxonomic identification of gut bacterial isolates of FAW according to their similarity to *GyrB* sequences.**

| Isolated code | NCBI-GenBank accession number | Phylogenetic affiliation |
|---|---|---|
| CYL2C1 | OR592215 | *Enterococcus mundtii* |
| CLL1C1 | OR592216 | *Enterococcus mundtii* |
| RLL2C5 | OR592217 | *Enterobacter huaxiensis* |
| CE1C3 | OR592218 | *Leclercia adecarboxylata* |
| CE1C6 | OR592219 | *Bacillus safensis* |

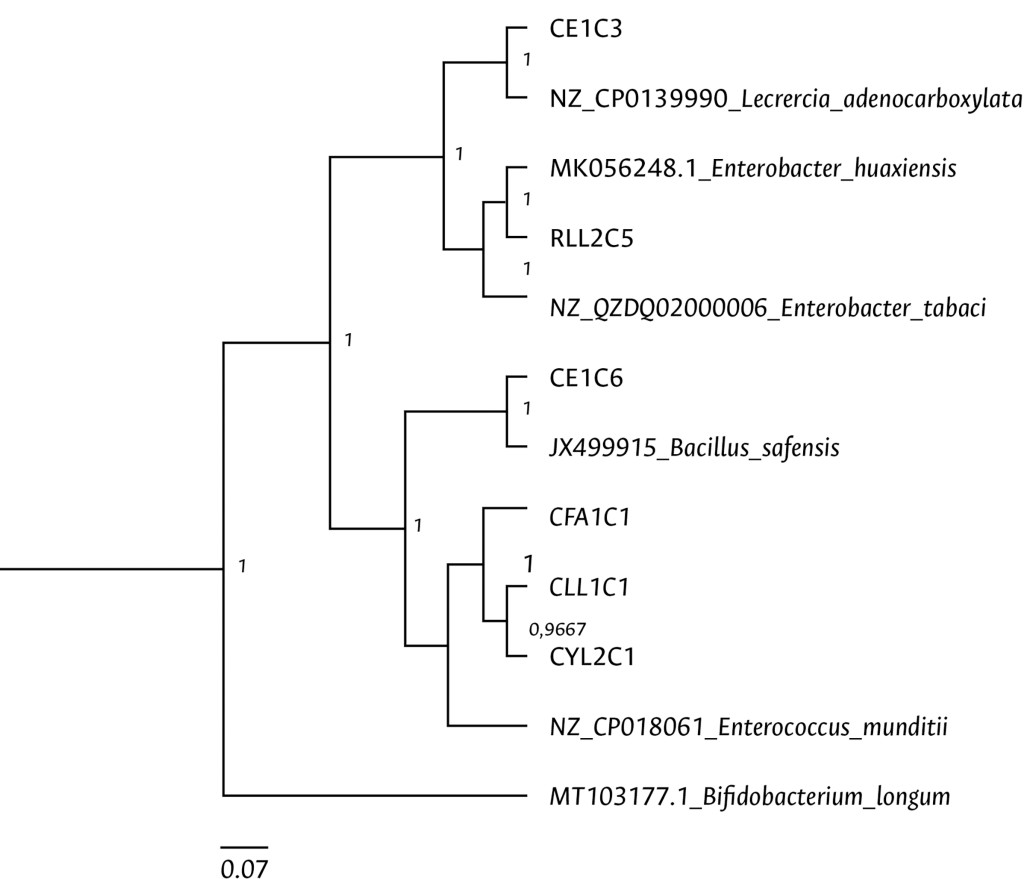

**Figure 6 Bayesian inference dendrogram for partial sequences of the gyrB gene of bacteria isolated from FAW gut samples.** General time reversible (GTR) substitution model with gamma distribution in the Beast program was used for the dendrogram construction. Bacterial isolates are represented with the assigned codes. The GenBank sequences are marked with the accession and identity code. The numbers in the nodes correspond to the posterior probability.

The genera *Bacillus* and *Staphylococcus* were also present in both strains. *Pediococcus, Cellulomonas, Paenibacillus, Lysinibacillus, Terribacillus, Curtobacterium*, and *Leclercia* were only found in the corn strain, while the *Enterobacter, Ca22rnobacterium*, and *Planococcus* genera were recovered from samples of the rice strain (Table 2).

## DISCUSSION

### Comparisons of FAW strains microbiota

Changes in gut microbiota due to host plant adaptation have been widely observed in several Lepidopteran species, particularly if larvae are fed directly from the leaves of the host plant (*Mason et al., 2020*; *Lv et al., 2021*; *Guo et al., 2022*; *Han et al., 2023*; *Jeon et al., 2023*; *Liu et al., 2023*). FAW microbiota diversity is greater in field-collected samples compared to laboratory samples (*Mason et al., 2020*; *Xu et al., 2022*; *Jeon et al., 2023*; *Han et al., 2023*; *Zheng et al., 2023*). The diet is a critical factor in modifying the composition and structure of the microbial community of FAW larvae (*Yuning et al., 2022*; *Wang et al., 2023*). In fact, FAW gut microbiota increases from gamma-irradiated leaves and an artificial diet to greenhouses corn leaves and field leaves (*Mason et al., 2020*). *Fu et al. (2023)* and *Li et al. (2022)* analyzed the microbiota of FAW corn populations through their life cycle. They also studied the microbiota effect on FAW reproduction. *Li et al. (2022)* compared the microbiota between artificial diet and corn leaves on the FAW corn strain. Both studies support our findings, as microbiota diversity was found to be higher in FAW eggs and larvae, and the microbiota composition was almost the same as what we reported. *Guo et al. (2022)* investigated the host plant preference, fitness costs, and differences in detoxification gene expression and microbiome composition between two *S. frugiperda* strains fed on different crop plant diets. However, they emphasized the response of the insect to host preference, finding phenotypic plasticity and P450 genes upregulated in CS. Our study provides additional information on CS and RS of FAW microbiota, as we analyzed their life cycle and successfully detected endosymbionts in eggs, pupae, and adults.

Comparisons of microbiota composition between FAW strains showed minor differences between CS and RS of FAW. However, our correlation analysis revealed that specific bacterial genera were associated with CS or RS of FAW. The genera *Colidextribacter*, *Pelomonas*, *Weissella*, and *Arsenophonus* were associated with CS, while *Leuconostoc*, A2, *Klebsiella*, *Lachnoclostridium*, *Spiroplasma*, and *Mucispirilum* were associated with RS. This suggests that FAW strains differ in some bacterial genera according to the host plant (*Han et al., 2023*). The low variation in FAW strains microbiota might be explained by the use of colony eggs, larvae, pupae, and adults. *Mason et al. (2020)* found that FAW gut bacterial abundance increased when larvae were fed with gamma-irradiated corn leaves and decreased when fed an artificial diet. *Han et al. (2023)* compared the microbiota from larvae feed until the fifth instar on several plants. They found that the diversity was higher in rice, followed by honeysuckle leaves, corn, wheat, Chinese yam, and finally, honeysuckle flowers. The most abundant genera were *Enterococcus*, *Staphylococcus*, and *Glutamicibacter* in CS and *Enterococcus*, *Erysiperlatoclostridium*, and *Klebsiella* in RS. Additionally, *Li et al. (2022)* compared the microbiota of CS during all developmental stages, finding more diverse values in field-collected samples than in laboratory samples. In a study conducted by *Jones et al. (2019)*, bacterial communities were analyzed in the foregut and midgut of corn earworms and FAW. They found that the gut microbiota is significantly influenced by the host plant.

However, other factors such as insect physiology, gut region, and local environmental factors can also contribute to the variation in the microbiome. In our research, we utilized insects fed an artificial diet. Though these insects were the F1 generation descended from wild insects, the change in their diet and living conditions could have affected the outcome of the study. The results obtained here could reflect the permanent FAW microbiota, but it's important to note that much of the microbiota found in a species can be transient. Nonetheless, the information obtained from this study is highly valuable as it provides useful insights into the bacterial composition of CS and RS of FAW. This information can be used to develop targeted strategies for managing this pest.

## FAW strains endosymbionts

In this study, we identified the endosymbionts *Arsenophonus* and *Spiroplasma* in CS and RS, respectively. *Castañeda-Molina et al. (2023)* also detected the genus *Arsenophonus* in CS in a work carried out concurrently with ours. However, our work further demonstrated the presence of this bacterium in Colombian FAW populations using an NGS approach. *Castañeda-Molina et al. (2023)* used specific primers to detect several endosymbionts, including *Arsenophonus* sp., *Spiroplasma* sp., *Cardinium* sp., and *Wolbachia* sp. Our work added more information about FAW microbiota since we detected *Arsenophonus* sp. in eggs, pupae, and adults and *Spiroplasma* sp. in eggs, larvae, pupae, and adults. *Dumas et al. (2015)* failed to detect the endosymbiont *Wolbachia* sp. in FAW. Here, this endosymbiont was also not identified by using metataxonomic analysis. However, *Schlum et al. (2021)* identified short fragments (<100 bp) matching the RefSeq *Wolbachia* in this pest. Additional studies are needed to confirm the presence of this endosymbiont in FAW. Other organs and specific primers should be examined to support the findings. Regarding the endosymbionts reported in this study, *Arsenophonus* in CS and *Spiroplasma* in RS, the results obtained are promising for FAW biological control. *Taylor et al. (2011)* found that *Arsenophonus nasoniae*, a gammaproteobacterium present in *Nasonia vitripennis*, kills a significant number of male embryos produced by an infected female wasp. On the other hand, *Spiroplasma* is a maternally transmitted bacterium associated with several species of flies and moths. This genus has the ability to cause male killing in *Drosophila melanogaster* (*Anbutsu & Fukatsu, 2011*). *Spiroplasma* has also been detected in Lepidoptera species, such as *Ostrinia zaguliaevi*, where it causes male killing (*Tabata et al., 2011*). Furthermore, in *Galleria mellonela*, the bacterium has been found to reduce larvae viability and induce male mortality (*Dowell, Basham & McCoy, 1981*).

*Fu et al. (2023)* removed the FAW gut microbiota using antibiotics and they found that egg hatching in the treated group was significantly reduced as compared to the control group. They also observed abnormal embryogenesis from FAW eggs. The researchers argue that antibiotics affect the reproductive capacity of FAW, especially in males. Further studies are needed to determine if microbiota plays a role in the reproduction of FAW strains. These studies should involve detecting endosymbionts in FAW and testing the effects of antibiotic treatments on their reproductive capacity.

## The dynamics of the microbial community throughout the life cycle of fall armyworm

FAW undergoes a complete metamorphosis in which a radical anatomical restructuring of the insect occurs, accompanied by functional and lifestyle changes (*Chen et al., 2016*). These changes can severely impact the composition of the microbiota during the different stages of its biological cycle (*Paniagua et al., 2018*; *Li et al., 2022*; *Fu et al., 2023*; *Lü et al., 2023*). This study investigated the composition of microbial communities throughout the biological cycle of FAW corn and rice strains using high-throughput sequencing.

The alpha diversity of the bacteria significantly changed during FAW's different development stages. Eggs from both CS and RS had the highest diversity and richness of bacterial species, while the pupae had the lowest. These findings are consistent with previous studies on FAW corn strains by *Fu et al. (2023)* and *Li et al. (2022)*, as well as on *S. littoralis* by *Chen et al. (2016)*. However, they differ from the results reported in *S. exigua*, where the greatest diversity in microbiota was found in the pupae stage (*Gao et al., 2019*).

Microbiota diversity in FAW adults exhibits a differential behavior that seems to be associated with sex. Males tend to display greater diversity than females. The Shannon, Simpson, and Chao1 indexes indicate that diversity is higher in FAW eggs and males compared to females. A study by *Fu et al. (2023)* also supports this observation, showing that eggs had the highest diversity, and female microbiota were less diverse than males (based on the Chao index, but the opposite results were found from the Shannon Index). On the other hand, in *S. littoralis*, females have the most diverse bacterial community compared to the other developmental stages (*Chen et al., 2016*).

Our findings reveal that Firmicutes and Proteobacteria were the dominant phyla in the life cycle of the FAW, which is in line with previous research conducted on FAW and other Lepidoptera species (*Chen et al., 2016*; *Gao et al., 2019*; *Gichuhi et al., 2020*; *González-Serrano et al., 2020*; *Hammer, McMillan & Fierer, 2014*; *Wang et al., 2023*; *Jeon et al., 2023*; *Fu et al., 2023*; *Zhou, Chen & Wang, 2022*). Although Firmicutes were the most commonly found bacteria in the gut microbiota of all evaluated FAW stages, Proteobacteria and Bacteroidetes were also present in high abundance in the eggs. Similar results were obtained in the corn strain of FAW by *Fu et al. (2023)*, *Li et al. (2022)*, *Gao et al. (2019)* in *S. exigua*, and *Chen et al. (2016)* in *S. littoralis*.

FAW is recognized as a polyphagous insect; its larval stages ingest large amounts of plant materials and other potentially harmful microbes associated with their food. In contrast, the adults will require energy for essential activities such as migration and reproduction (*Guo et al., 2022*; *Fu et al., 2023*). Firmicutes and Proteobacteria species are involved in the degradation of complex dietary molecules (cellulose and hemicellulose) and the metabolism of amino acids, demonstrating the importance of these bacteria in insects (*Fonknechten et al., 2010*; *Suen et al., 2010*; *Fu et al., 2023*) also Firmicutes and Proteobacteria may play important auxiliary roles in the growth and development of FAW larvae and adults, as well as in fighting pathogens (*Fu et al., 2023*). In this study, as well as in *Li et al. (2022)*, young larvae exhibited higher microbiota diversity than older larvae;

they explain that since the food intake of the late larval instars (L4–L6) was significantly increased compared with that of early larval instars (L1–L3) and the body size grew faster, the changes in the gut microbiota might be associated with the growth and development of the host insects, which was consistent with previous reports in *Bombyx mori*. According to *Li et al. (2022)*, different investigations have shown that early larval stages are more sensitive to environmental changes, which are related to their body sizes and the development of their immune systems. For this reason, microbiota diversity may be higher in younger FAW larvae.

A total of 81 genera were identified, with *Enterococcus* being the predominant genus in larvae, pupae of both strains, as well as in females of CS. The proportion of richness was 5% minor in RS for this genus. Also, the genus *Illeobacterium* was predominant in both strains. Other commonly found genera found in eggs of both strains and CS males included *Dobosiella, Turicibacter Burkholderia, Bifidobacterium, Lactobacillus*, and *Ralstonia*. *Carnobacterium* exhibited a high abundancy in pupae of both strains. *Illeobacterium, Ralstonia*, and *Burkholderia* have similar abundancies in all stages of development in both strains.

*Enterococcus* is a genus of lactic acid bacteria with high metabolic and adaptive versatility (*Ramsey, Hartke & Huycke, 2014*). This genus is commonly found in the gut of several Lepidoptera species (*Paniagua et al., 2018*; *Mason et al., 2020*; *Fu et al., 2023*; *Jeon et al., 2023*; *Han et al., 2023*), and it has been reported as the dominant taxon in the gut Lepidoptera such as *S. littoralis* (*Chen et al., 2016*), *S. exigua* (*Gao et al., 2019*), *Manduca sexta* (*Brinkmann, Martens & Tebbe, 2008*), *Helicoverpa armigera* (*Xiang et al., 2006*), *Lymantria dispar* (*Broderick et al., 2004*), and in *S. frugiperda* (*Higuita-Palacio et al., 2021*; *Li et al., 2022*; *Lü et al., 2023*; *Fu et al., 2023*; *Jeon et al., 2023*; *Han et al., 2023*; *Oliveira & Cônsoli, 2023*). The persistence of *Enterococcus* throughout the life cycle of FAW suggests that this bacterium possesses efficient adaptive mechanisms that allow it to remain through different FAW developmental stages, and therefore, it might be vertically transmitted. *Chen et al. (2021)* proposed that *Enterococcus* plays a crucial role in the carbohydrate transport, metabolism, and energy production in the gut of FAW larvae. This genus of bacteria has been found to enhance the growth rate of FAW larvae even when they are fed suboptimal diets (*Chen et al., 2022*). Furthermore, *Enterococcus* has been associated with pesticide degradation and resistance in FAW (*Almeida et al., 2017*; *Gomes, Omoto & Cônsoli, 2020*).

Female moths lay their eggs on host plants, which are then covered by their scales and microbiota (*Rwomushana, 2019*). Since the eggs are exposed to the environment, female scales protect them from abiotic and biotic pressures. The microbiota also protects the eggs from other harmful bacteria, viruses, and fungus colonization. Due to the bacterial load from females and exposure to environmental conditions, eggs of FAW have a higher diversity than other developmental stages of the insect (*Chen et al., 2016*). *Chen et al. (2016)* explained that in *S. littoralis*, larvae consume all the chorion and its microbiota after egg eclosion. Thus, microbiota can easily be vertically transmitted. We found that microbiota diversity is highest in eggs and decreases in larvae, pupae, and adults, particularly in females. These results are similar to the findings in FAW by *Fu et al. (2023)*

and in *S. littoralis* by *Chen et al. (2016)*. In FAW, the gut microbiota diversity is lower in older larvae than in younger ones because bacterial competition is likely to occur in the gut. Most of the microbiota identified in this study are composed of the genus *Enterococcus*. Studies made in other moths, such as *Manduca sexta* and *Helicoverpa armigera*, showed that lysozyme is produced from the beginning of the insect metamorphosis (*Russell & Dunn, 1991*; *Zhang et al., 2009*) and this enzyme catalyzes the hydrolysis of bacterial walls (N-acetylmuramic and N-acetyl-D-glucosamine), generating bacterial lysis (*Ragland & Criss, 2017*). Therefore, interspecies bacterial competition and several gut enzymes can act as selection pressure against insect microbiota. For this reason, bacteria genera found in this study are almost the same compared to other Lepidoptera species where the genus *Enterococcus* was the most predominantly found from eggs to adults (*Chen et al., 2016*; *Gao et al., 2019*; *Higuita-Palacio et al., 2021*; *Castañeda-Molina et al., 2023*; *Oliveira & Cônsoli, 2023*; *Paniagua et al., 2018*; *Fu et al., 2023*).

*Enterococcus* and *Ileibacterium* were identified as the microbial core in the samples of FAW CS and RS strains in this work. *Fu et al. (2023)* explained that *Enterococcus* can degrade alkaloids and latex and has a putative role in detoxifying plant toxins. Additionally, *Enterobacter* contributes to the synthesis of vitamins and pheromones, the degradation of plant compounds, and the process of nitrogen fixation. The higher abundances of *Enterococcus* and *Enterobacter* at the larval, pupal, and adult stages imply that they may contribute to FAW nutrient absorption (*Fu et al., 2023*; *Jeon et al., 2023*), and for this reason, they might be mostly found in females than in males as females are relevant for insect´s segregation through their progeny.

*Ileibacterium, Dubosiella, Turicibacter, Ralstonia*, and *Burkholderia* were the most abundant genera found in eggs and were also found in L1-3 larvae and males. *Ileibacterium* and *Dubosiella* are two recently described genera of the Erysipelotrichaceae family that comprise anaerobic or aerotolerant bacteria that grow fermentatively on several carbohydrates, including hexoses, pentoses, and disaccharides (*Cox et al., 2017*). Bacteria of the Erysipelotrichaceae family have been mainly isolated from mammals' guts (*Dimet-Wiley et al., 2022*; *Kaakoush, 2015*) and insects' guts (*Juottonen et al., 2022*) and have also been detected in the gut of FAW larvae in previous studies made in Colombia (*Castañeda-Molina et al., 2023*; *Higuita-Palacio et al., 2021*) and Brazil (*Oliveira & Cônsoli, 2023*). On the other hand, the genus *Burkholderia*, which was abundantly found in males, is implicated in pesticide degradation in several insect species (*Kikuchi et al., 2012*; *Tago et al., 2015*). Females exhibited lower microbial diversity and richness than males, and the microbiota of males was similar to the microbiota detected in eggs. Differences in FAW male and female microbiota found in this study might suggest a vertical transmission during copulation, as demonstrated in *S. exigua* by *Gao et al. (2019)*.

In order to support the culture-independent microbiota analysis, a culture-dependent strategy was also carried out in this study. The analysis of 16S rDNA partial sequences and the *gyr*B gene of bacterial isolates revealed that the community of culturable bacteria was mainly composed of members of Firmicutes, Proteobacteria, and Actinobacteria phyla. Moreover, similar to the results found with the high-throughput sequencing protocol, *Enterococcus* was the predominant genus in all development stages and strains of FAW.

The analysis of partial sequencing of 16S rDNA allowed the identification of the species *E. mundtii*, *E. casseliflavus*, and *E. haemoperoxidus*. Three isolates identified as *E. mundtii* were further confirmed by *gyrB* gene sequencing. *E. mundtii* is a lactic acid bacterium with probiotic and biofilm properties. The species produces bacteriocins that can inhibit the growth of other bacteria (*Magni et al., 2012*). These characteristics may explain their predominance in the FAW gut. In *S. littoralis*, strains of *E. mundtii* can produce mundticin KS that inhibits some bacteria growth (*Shao et al., 2017*). In *G. mellonella*, *E. mundtii* can be found in all developmental stages and seems to play an important role in preventing the establishment of the genera *Serratia* and *Staphylococcus* bacteria (*Johnston & Rolff, 2015*). *Chen et al. (2016)* found that *E. mundtii* persists during the life cycle of *S. littoralis*, serving as a metabolically active bacteria in the gut of this pest (*Shao, Arias-Cordero & Boland, 2013*). In this work, *E. mundtii* was also present during the FAW's entire life cycle, suggesting that the species is very competitive in this insect. Our results demonstrate that this species can survive in this host better than other bacteria. *Almeida et al. (2017)* observed that *E. mundtii* can tolerate and degrade lambda-cyhalothrin, spinosad, lufenuron, and ethyl chlorpyrifos. *E. mundtii* was also detected by *Castañeda-Molina et al. (2023)* in the FAW corn strain, finding that it is capable of resisting antibiotic treatment and persists in the presence of several *Bt* endotoxins. Our results suggest that this bacterium can potentially be used in further studies based on biological control of FAW, as it has been detected in the majority of studies made on this pest (*Li et al., 2022*; *Fu et al., 2023*; *Jeon et al., 2023*).

Using the culture-dependent approach, *Bacillus* and *Staphylococcus* were isolated from FAW corn and rice strains, in addition to *Enterococcus* species. Moreover, the genera *Pediococcus, Cellulomonas, Paenibacillus, Lysinibacillus, Terribacillus, Curtobacterium*, and *Leclercia* were mainly found in CS, while the genera *Enterobacter, Carnobacterium*, and *Planococcus* in RS. *Han et al. (2023)* observed that the most abundant genera in FAW were *Enterococcus, Staphylococcus*, and *Glutamicibacter* in CS and *Enterococcus, Erysiperlatoclostridium*, and *Klebsiella* in RS. The differences between our study and theirs may be due to the fact that colony FAW insects were examined in our investigation, whereas *Han et al. (2023)* used wild insects. Two members of the Proteobacteria phylum were isolated and identified as *Leclercia adenocarboxylata* and *Enterobacter tabacci*. The identification of *L. adenocarboxylata* was further confirmed through sequencing of the *gyr*B gene. However, the isolate initially identified as *E. tabaci* was reclassified as *E. huaxiensis* using the *gyr*B gene. While the 16S rRNA gene is a useful marker in the taxonomic classification of bacteria, it has limitations in identifying closely related species (*Gunther et al., 2011*), such as those found in the *Enterobacter* genus. The enzyme gyrase subunit B gene has a higher evolution rate (*Watanabe et al., 2001*), making it more suitable for identifying species like *L. adenocarboxylata* and *E. huaxiensis* that belong to the Enterobacteriaceae family. These bacteria are ubiquitous in the digestive tract of different species of animals (*Brenner et al., 2005*). *L. adenocarboxylata* has a potential use against *Leptinotarsa decemlineata* given its insecticidal activity (*Muratoglu et al., 2009*). In *S. frugiperda*, *Almeida et al. (2017)* found that *L. adenocarboxylata* was resistant to

lambda-cyhalothrin, deltamethrin, spinosad, lufenuron, and ethyl chlorpyrphos in FAW populations, showing that this species can degrade insecticides under *in vitro* conditions.

According to *Paniagua et al. (2018)*, symbionts found in Lepidoptera, such as *Bacillus*, *Enterococcus*, *Staphylococcus*, and *Enterobacter*, are among the most common found in this order, as they have been detected in more than 70% of its species. This study found these genera using both culture-dependent techniques and high-throughput sequencing. *Carnobacterium* was another genus identified in this work. The isolate had a 16S rDNA sequence with 96.6% similarity with *C. maltaromaticum* NR_044710.2 in GenBank. Some *C. maltaromaticum* strains can produce up to three bacteriocins that can inhibit the growth of bacteria such as *E. faecalis*, *E. faecium*, *Pediococcusacidilactici*, *C. divergens*, *Lactococcus lactis*, *Lactobacillus curvatus*, *Lactobacillus casei*, *Leuconostocgelidum*, *S. aureus*, and *Clostridium botulinum* (*González et al., 2013*). *Xia et al. (2017)* identified *C. maltaromaticum* in *Plutella xylostella* larvae, pupae, and adults. This species plays a role in the degradation of complex carbohydrates in this pest's intestine. Here, *C. maltaromaticum* was isolated from L1-3 larvae, and the NGS analysis showed its relative abundance was 17% in L1-3 larvae, 3% in L4-6 larvae, 32% in pupae, and 20% in males. Another lactic acid bacterium identified in this study was *Pediococcus pentosaceus*. In *Tenebrio molitor*, *Pediococcus* has a beneficial effect on insect development (*Lecocq et al., 2021*).

Finally, *Cellulomonas* and *Curtobacterium* were two Actinobacteria genera identified in FAW. Both have been reported as plant growth-promoting bacteria (*Ahmed et al., 2014*; *Patel et al., 2022*). ASVs of these bacteria were found by NGS, although at very low relative abundances.

This study is the first to compare the microbiota of FAW corn and rice strains across their life cycle, and it found some differences in their microbiota composition. *Ileibacterium*, *Dubosiella*, *Ralstonia*, *Turicibacter*, and *Bifidobacterium* were differentially abundant in eggs, while *Enterococcus*, *Burkholderia*, and *Carnobacterium* were differentially abundant in adult females, adult males, and pupae, respectively. *Leuconostoc*, *Klebsiella*, *Lachnoclostridium*, *Spiroplasma*, and *Mucispirilum* are more abundant in RS, while *Colidextribacter*, *Pelomonas*, *Weissella*, and *Arsenophonus* in CS. Endosymbionts *Arsenophonus* and *Spiroplasma* found in this work are involved in the embryo and male-killing in another Lepidoptera. The outcomes obtained here provide further information on FAW microbiota that has not been documented elsewhere. Managing this pest requires combining different strategies to control it in nature, particularly by considering FAW strains, as they have shown differences in their evolutionary biology and insecticide resistance.

### Funding

This research was funded by Ministerio de Ciencias y Tecnología (2018–2023) under the Project entitled: "Bioprospección de la microbiota asociada a insectos plaga de cultivos de interés agrícola en Colombia: *Spodoptera frugiperda* (biotipos maíz y arroz) y trips del

aguacate para el desarrollo de alternativas de manejo de su control, ID Hermes 42409 Universidad Nacional de Colombia. MinCiencias 80740-146-2019" Convocatoria para proyectos de ciencia, tecnología e innovación y su contribución a los retos de país–808. The funders had no role in study design, data collection and analysis, decision to publish, or preparation of the manuscript.

## Grant Disclosures
The following grant information was disclosed by the authors:
Ministerio de Ciencias y Tecnología (2018–2023).
Universidad Nacional de Colombia.

## Competing Interests
The authors declare that they have no competing interests.

## Author Contributions
- Sandra María Marulanda-Moreno conceived and designed the experiments, performed the experiments, analyzed the data, prepared figures and/or tables, authored or reviewed drafts of the article, and approved the final draft.
- Clara Inés Saldamando-Benjumea conceived and designed the experiments, analyzed the data, authored or reviewed drafts of the article, and approved the final draft.
- Rafael Vivero Gomez analyzed the data, prepared figures and/or tables, authored or reviewed drafts of the article, analysis tools, and approved the final draft.
- Gloria Cadavid-Restrepo conceived and designed the experiments, analyzed the data, authored or reviewed drafts of the article, and approved the final draft.
- Claudia Ximena Moreno-Herrera conceived and designed the experiments, analyzed the data, prepared figures and/or tables, authored or reviewed drafts of the article, and approved the final draft.

## Field Study Permissions
The following information was supplied relating to field study approvals (*i.e.*, approving body and any reference numbers):

The collection of the larvae and genetic access was provided by ANLA (Autoridad Nacional De Licencias Ambientales to Universidad Nacional de Colombia. Permiso marco de recolección de especímenes silvestres, resolución 0255, 14/03/2014 (artículo 3).

## DNA Deposition
The following information was supplied regarding the deposition of DNA sequences:

The sequences of 16S rDNA are available at GenBank: OR546283 to OR546319, and for gyrB gene are OR592215 to OR592219.

## Data Availability
The data is available at Figshare: Marulanda Moreno, Sandra María; Vivero, Rafael Jose; Moreno-Herrera, Claudia Ximena; Ester Cadavid-Restrepo, Gloria; Ines Saldamando-

Benjumea, Clara (2023). *Spodoptera frugiperda* strain rice and corn Microbiome Dataset. figshare. Dataset. https://doi.org/10.6084/m9.figshare.24192972.v1.

## Supplemental Information

Supplemental information for this article can be found online at http://dx.doi.org/10.7717/peerj.17087#supplemental-information.

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
