# Peer review of "Comparative analysis of Spodoptera frugiperda (J. E. Smith) (Lepidoptera, Noctuidae) corn and rice strains microbiota revealed minor changes across life cycle and strain endosymbiont association"

_PeerJ, doi:10.7717/peerj.17087_

## Round 0.1 · original submission · Major Revisions

Three experts have assessed your manuscript and raised several concerns about the form and structure of the manuscript, along with the poor English usage. There are also concerns related to numerous self-citations that should be avoided, as they do not add much to the background/discussion of this work.

**Language Note:** The Academic Editor has identified that the English language must be improved. PeerJ can provide language editing services - please contact us at [email protected] for pricing (be sure to provide your manuscript number and title). Alternatively, you should make your own arrangements to improve the language quality and provide details in your response letter. – PeerJ Staff

Reviewer 1 ·

Basic reporting

The manuscript meets PeerJ standards and covers criteria of writing scientific publications. All sections are well covered. The English language is clear and the paper is well written.

Experimental design

The research is original. I do not have serious concerns about the general approach, the overall structure, and the originality of the research.

Validity of the findings

I commend the authors for several strengths of their work. In this manuscript, Marulanda-Moreno et al., demonstrated differences and similarities in abundance and richness between gut microbiota of two fall armyworm strains (corn and rice stains). The results are important for paving the way toward characterization of corn and rice strains microbiota to develop new strategies for pest control.

Additional comments

Considering several strengths of their work, the manuscript requires minor revision and can be further strengthened by addressing reviewer’s comments. My comments are detailed in the reviewer's remarks to the authors. It briefly includes, revising the manuscript to correct several typos and grammatical errors, make sure the title is well written and represent the outcomes, all figures should be reproduced due to the low quality and inappropriate font size, and the discussion section should be abridged and deepened around findings and related work.
* * *
Reviewer's remarks to the authors.
Article title: “Microbiota Spodoptera frugiperda (J.E. Smith) (Lepidoptera, Noctuidae) corn and rice strains comparison through their developmental stages show endosymbiont association and no major host plant differentiation”
Journal: PeerJ

Dear Authors,

I appreciate the opportunity to review your interesting manuscript. I enjoyed reading the manuscript and I commend you for several strengths of your work. The study demonstrated differences and similarities in abundance and richness between gut microbiota of two fall armyworm strains (corn and rice stains). The results are important for paving the way toward characterization of corn and rice strains microbiota to develop new strategies for pest control. I do not have serious concerns about the general approach, the overall structure, and the originality of the research. However, I have some remarks to the authors that might improve the manuscript.

Comments:

With the strengths of your work, I believe the paper could be further strengthened by considering the following comments.

The title has some grammatical errors. I believe that the title should be revised carefully or consider the following suggestion:

Comparative analysis of Spodoptera frugiperda (J.E. Smith) (Lepidoptera, Noctuidae) corn and rice strains gut microbiota revealed changes among developmental stages and strain endosymbiont association with minor strain differences”

The paper is well written but there are some typos and grammatical errors. For instance:

Line 31: “Illumina MySeq” to “MiSeq”

Line 100: delete “here”

Line 108/109: “the following insecticides” to “insecticides from various categories such as …”

Line 116: delete “also”

Line 284: add “,” after “males”.

Line 150: change “deposited” to “placed”.

Line 151: delete “Also”.

Line 358: change “separated” to “another”.

In several places, there is inappropriate use of “by using.” (See lines, 164, 211, …).

Please revise the entire manuscript for such minor mistakes.

In the material and methods: Line 140-143: Clarify which generation is used for the analysis.

Most figures have low resolution and low quality. Please reproduce all figures with higher quality and appropriate font size.

Figure 1 and table 1 are not essential to be in the paper context. It should be moved to supplementary materials.

Figure 2: add the title of x axis, no need to mention statistic or F value, add P value into the figure, use stars (* , **, ...) in panels A-C, and letter of HSD significancy (a, b, c ..) in panels D-F to show significant differences, and define that as a footnote under the figure.

Figure 3: add the percentage of contribution of each dimension (x axis and y axis) in panel C. Also please look at PC3 and PC4 to see If any may have high contribution or separation between groups. Keep it consistent, either add information of PCA (Principal Component Analysis) statistical analysis to all panels or not to all panels. in the context, what about the PREMANOVA results of other panels or other dimensions. Add the levels of confidence that you are using to draw a grouping of points in figure 3.

The numbering system of supplementary figures should match the system used in the context.

In the discussion section, please try to abridge and deepen your discussion, especially for interpretation of your results without repeating the results (e.g., Line 466-469, 470-474, 487-489 … it is noticeable in several other places). Also, please give more attention to revise English and any grammatical errors in this section.

It was an immense pleasure to accept the invitation for peer review of your manuscript and hope that these comments will help to improve the manuscript.

Reviewer 2 ·

Basic reporting

Title: I consider that title is very long. The title should be shorter and more concise.
English: the English language is good. No have comments.
References: The bibliographic references are related to the research topic, it is necessary to review the format of the bibliography (line 563-954).
The article has an orderly structure and the most relevant points are mentioned in each section. The legends of the tables and figures are fine, only some spelling errors need to be corrected. The figures are interesting and easy for the reader to understand, however, the quality needs to be improved. Figure 8 has the name Carnobacterium underlined in red.

Experimental design

The objective is clear, however, I consider that the association made between the insect microbiota and the importance of establishing the species of corn and rice crops is confusing.
The methodology is very well planned and developed. Regarding this section of the article I have no comments. Only review comments on line 135-136, line 214 (8% polyacrylamide).

Validity of the findings

The methodology used contributes to the work being complete and concise. The statistical analyzes used and the bioinformatics tools are indicated to answer your research question.The results are well explained, and cover the most important points that were found in the investigation.

Additional comments

1. Why is it that Firmicutes and Proteobacteria are found in greater proportions in females than in males?
2. Why does the microbiota vary between morphological stages?
3. In the discussion, more emphasis should be placed on the importance of knowing the microbiota of this insect and its biological control in corn and rice fields.
4. The beginning of the discussion (line 341-406) should be summarized, I consider that the information is repetitive and does not conclude anything specific.
5. The discussion seems more like a summary of results than a discussion, this section should be revised.

Annotated reviews are not available for download in order to protect the identity of reviewers who chose to remain anonymous.

Reviewer 3 ·

Basic reporting

The study submittedThe submitted study aims to analyze variations in the gut microbiota composition of Spodoptera frugiperda based on strain genotype and life stages. Unfortunately, the paper is marred by poor English, convoluted sentences, and numerous formal errors throughout the text. Moreover, the authors overlook a substantial body of existing literature on the topic, failing to cite or discuss it at any point. Instead, they rely on self-citations unrelated to the study.
Furthermore, a critical flaw in the experimental design is highlighted in the "experimental design" section, and regrettably, the authors do not address this issue anywhere in the text.
I recommend rejecting the manuscript due to its failure to meet the standard quality for publication. However, I am open to reconsidering if the authors commit to a comprehensive rewrite. This revision should encompass improvements in English writing, a thorough scientific analysis of their results within the context of previously published literature, and a reevaluation of the experimental design.
SOME EXAMPLES OF POORLY WRITTEN SENTENCES AND FORMAL ERROR (SPECIES NAMES, Etc)
Line 66: “…as a secondary pest in crops of rice”
Line 70 – 74. The two sentences separated by “and” have nothing in common. I do not understand the structure of this reasoning.
Line 90: “Insect-associated microorganisms play an essential role in their diversification and evolutionary success on the hosts”
Line 91: “For example, some genera of intestinal bacteria are involved in the nutritional process of degradation and ingestion of insects”. Are intestinal bacteria involved in the ingestion of insects?
Line 91-97: very large sentence.
Line 99: “A recent study made in Colombia demonstrated the presence of two species of the genus Enterococcus in FAW corn strain gut microbiota that were resistant to Bt endotoxins and antibiotics under laboratory conditions and the presence of a gonad endosymbiont also detected here.” Are Enterococcus strains resistant to Bt?
Line 133: “The larvae were fed with corn and rice leaves until they hatched.” Do larvae hatch?
And so on…
All along the text: The name of the species is Spodoptera littoralis, no litoralis with a single n.
Figure 2: T-test statistics has a name, t
OVERLOOKED PUBLISHED STUDIES
1.SEMINAL WORK THAT I CANNOT UNDERSTAND HOW COULD IT BE OVERLOOKED
• Jones et al., 2019. Host plant and population source drive diversity of microbial gut communities in two polyphagous insects. Authors tested exactly the same hypothesis than the submitted paper, in an unbiased manner (and with results contrasting those presented in the submitted study)
• Mason et al 2020. Diet influences proliferation and stability of gut bacterial populations in herbivorous lepidopteran larvae. Authors checked in an unbiased manner the bacteria community of S. frugiperda along larval stages
• Fu et al., 2023. Composition and diversity of gut microbiota across developmental stages of Spodoptera frugiperda and its effect on the reproduction. Another study not mentioned in the submitted paper but testing exactly the same hypothesis (differences across life stages)
• Guo et al., 2022. Insecticide Susceptibility and Mechanism of Spodoptera frugiperda on Different Host Plants. Another study not mentioned in the submitted paper but testing exactly the same hypothesis (differences between strains)
• Acevedo et al., 2017. Fall Armyworm-Associated Gut Bacteria Modulate Plant Defense Responses. Authors identified Enterobacter strains involved in plant defense response
• Mason et al., 2022. Opposing Growth Responses of Lepidopteran Larvae to the Establishment of Gut Microbiota. Authors established the role of enterococcus and enterobacter strains help S. frugiperda in poor diet conditions
• Chen et al., 2022. Enterococcal symbionts of caterpillars facilitate the utilization of a suboptimal diet

2.ADDITIONAL SIMILAR WORKS THAT SHOULD BE DISCUSSED IN THE TEXT
• Liu et al., 2023. Investigation of the fall armyworm (Spodoptera frugiperda) gut microbiome and entomopathogenic fungus-induced pathobiome
• Jeon et aal., 2023. Spodoptera frugiperda (Lepidoptera: Noctuidae) Life Table Comparisons and Gut Microbiome Analysis Reared on Corn Varieties
• Xu et al., 2022. Effect of Diet on the Midgut Microbial Composition and Host Immunity of the Fall Armyworm, Spodoptera frugiperda
• Han et al., 2023. Effect of Different Host Plants on the Diversity of Gut Bacterial Communities of Spodoptera frugiperda (J. E. Smith, 1797)
• Zhu et al., 2022. Gut Bacterial Diversity and Community Structure of Spodoptera exigua (Lepidoptera: Noctuidae) in the Welsh Onion-producing Areas of North China
• Li et al., 2022. Fall Armyworm Gut Bacterial Diversity Associated with Different Developmental Stages, Environmental Habitats, and Diets
• Zheng et al., 2023. Comparative analysis of gut microbiota and immune genes linked with the immune system of wild and captive Spodoptera frugiperda (Lepidoptera: Noctuidae)
• Yuning et al. 2022. The bacterial and fungal communities of the larval midgut of Spodoptera frugiperda (Lepidoptera: Noctuidae) varied by feeding on two cruciferous vegetables
• Lv et al., 2021. Comparison of Gut Bacterial Communities of Fall Armyworm (Spodoptera frugiperda) Reared on Different Host Plants
• Mason et al., 2021. Effects of maize (Zea mays) genotypes and microbial sources in shaping fall armyworm (Spodoptera frugiperda) gut bacterial communities
• Gichuhi et al., 2020. Diversity of fall armyworm, Spodoptera frugiperda and their gut bacterial community in Kenya

WRONG CITATIONS OR SELF CITATIONS
• Cano-Calle, D. (2020). Caracterización Molecular de trips (Thysanoptera: Thripidae) procedentes de cultivos comerciales de aguacate (Persea americana Mill) del oriente antioqueño y estudio de la diversidad microbiana asociada. Universidad Nacional de Colombia. P 180. REASON: I have the feeling that citing a PhD thesis in Spanish for describing a PCR temperature cycle should be avoid, better to include the temperature cycle in the text.
• From line 366 to 370, arguments to describe reproductive isolation of S. frugiperda subspecies only derive from papers published by the group submitting the paper (a cited paper is even in Spanish), ignoring a vast literature produced worldwide. See for example:
o Groot et al 2008 Host strain specific sex pheromone variation in Spodoptera frugiperda;
o Dumas et al., 2015; Spodoptera frugiperda (Lepidoptera: Noctuidae) host-plant variants: two host strains or two distinct species?;
o Kost et al, 2016; A Z-linked sterility locus causes sexual abstinence in hybrid females and facilitates speciation in Spodoptera frugiperda;
o Cruz-Esteban et al., 2017; Calling Behavior, Copulation Time, and Reproductive Compatibility of Corn-Strain Fall Armyworm (Lepidoptera: Noctuidae) From Populations in Mexico.
o Tessnow et al., 2022.
o Patterns of genomic and allochronic strain divergence in the fall armyworm, Spodoptera frugiperda (J.E. Smith).
o Fiteni et al., 2022; Host-plant adaptation as a driver of incipient speciation in the fall armyworm (Spodoptera frugiperda).
• Oliveira NC, Rodrigues PAP, Cônsoli FL. 2021. Host-adapted strains of Spodoptera frugiperda hold and share a core microbial community across the western hemisphere. BioRxiv. This paper has been published in a peer-reviewd journal, citations should point to this final version (Oliveira NC, Rodrigues PAP, Cônsoli FL. Host-Adapted Strains of Spodoptera frugiperda Hold and Share a Core Microbial Community Across the Western Hemisphere. Microb Ecol. 2023)
• More than one cited paper is “in press”

Experimental design

One of the main hypotheses of the paper is to highlight differences in the gut microbiota between rice and corn strains. For this purpose, the authors collected wild insects, which were later grown in laboratory conditions under an artificial diet. The diet strongly shapes gut bacterial communities in Spodoptera species. The authors should take this important point into consideration when they conclude 'no major host plant differentiation.

(see for example
- Martinez solis et al., Influence of Diet, Sex, and Viral Infections on the Gut Microbiota Composition of Spodoptera exigua Caterpillars
- Mason et al 2020. Diet influences proliferation and stability of gut bacterial populations in herbivorous lepidopteran larvae.
• Xu et al., 2022. Effect of Diet on the Midgut Microbial Composition and Host Immunity of the Fall Armyworm, Spodoptera frugiperda
• Mason et al., 2021. Effects of maize (Zea mays) genotypes and microbial sources in shaping fall armyworm (Spodoptera frugiperda) gut bacterial communities
• Han et al., 2023. Effect of Different Host Plants on the Diversity of Gut Bacterial Communities of Spodoptera frugiperda (J. E. Smith, 1797)
and many others...)

Validity of the findings

I did not check this point because it is useless until all the flaws highlighted above are fixed

Additional comments

---

## Round 0.2 · accepted · Accept

The authors properly addressed the concerns raised by the three Reviewers. This version is suitable for publication.

Reviewer 2 ·

Basic reporting

Regarding the language, there was nothing to improve, only some editing comments which were taken care of.
The formatting of the references was improved, as well as the spelling errors.

Experimental design

The authors took into account each of the comments made to improve the article. Figures were improved and others were added to better understand the results.

Validity of the findings

I consider that the respective modifications have been made to this section. I have no further comments to improve it.

Additional comments

Taking into account the comments that were sent to the authors, each one of them was addressed, so I consider that the paper is ready to be published in this version.